# Farming for Pharming: Novel Hydroponic Process in Contained Environment for Efficient Pharma-Grade Production of Saffron

**DOI:** 10.3390/molecules27248972

**Published:** 2022-12-16

**Authors:** Luca Nardi, Giulio Metelli, Marco Garegnani, Maria Elena Villani, Silvia Massa, Elisabetta Bennici, Raffaele Lamanna, Marcello Catellani, Silvia Bisti, Maria Anna Maggi, Olivia C. Demurtas, Eugenio Benvenuto, Angiola Desiderio

**Affiliations:** 1ENEA, Italian National Agency for New Technologies, Energy and Sustainable Economic Development, Biotechnology and Agro-Industry Division, Casaccia Research Center, 00123 Rome, RM, Italy; 2DAER—Department of Aerospace Science and Technology, Politecnico of Milano, 20100 Milan, MI, Italy; 3ENEA, Italian National Agency for New Technologies, Energy and Sustainable Economic Development, Biotechnology and Agro-Industry Division, Trisaia Research Center, 75026 Rotondella, MT, Italy; 4National Institute of Biostructures and Biosystems (INBB), Viale Medaglie D’Oro 305, 00136 Rome, RM, Italy; 5Hortus Novus Srl, Via Campo Sportivo 2, 67050 Canistro, AQ, Italy

**Keywords:** saffron, hydroponic cultivation, pharma-grade production

## Abstract

Soilless cultivation of saffron (*Crocus sativus*) in a controlled environment represents an interesting alternative to field cultivation, in order to obtain a standardized high-quality product and to optimize yields. In particular, pharma-grade saffron is fundamental for therapeutic applications of this spice, whose efficacy has been demonstrated in the treatment of macular diseases, such as Age-related Macular Degeneration (AMD). In this work, a hydroponic cultivation system was developed, specifically designed to meet the needs of *C. sativus* plant. Various cultivation recipes, different in spectrum and intensity of lighting, temperature, photoperiod and irrigation, have been adopted to study their effect on saffron production. The experimentation involved the cultivation of corms from two subsequent farm years, to identify and validate the optimal conditions, both in terms of quantitative yield and as accumulation of bioactive metabolites, with particular reference to crocins and picrocrocin, which define the ‘pharma-grade’ quality of saffron. Through HPLC analysis and chromatography it was possible to identify the cultivation parameters suitable for the production of saffron with neuroprotective properties, evaluated by comparison with an ISO standard and the REPRON^®^ procedure. Furthermore, the biochemical characterization was completed through NMR and high-resolution mass spectrometry analyses of saffron extracts. The whole experimental framework allowed to establish an optimized protocol to produce pharma-grade saffron, allowing up to 3.2 g/m^2^ harvest (i.e., more than three times higher than field production in optimal conditions), which meets the standards of composition for the therapy of AMD.

## 1. Introduction

Saffron (*Crocus sativus* L.) belongs to the *Iridaceae* family and is vegetatively propagated through corms because cultivated plants are sterile auto-triploids. This spice is among the most expensive and appreciated foodstuffs in the world, in consideration of the single harvest per year, the limited areas naturally suitable for cultivation and the manual work needed to procure the dried stigmas [1,2]. Saffron is valuable not only for its organoleptic and nutraceutical properties as a food ingredient, but also for its efficacy in therapeutic treatments, attributed to three main bioactive compounds, crocins, picrocrocin and safranal [3,4,5]. In-field saffron yield is extremely variable, depending on agronomic, environmental and biological factors with an average of dry stigmas ranging from 1.6 to 2.4 kg/ha [6]. Higher yields have been obtained with elaborate agronomic measures and in particularly favorable environmental conditions reach values up to 10 kg/ha in Italy, with a planting density usually of 20–80 corms per square meter [7,8,9,10]. Due to increasing labor costs and climate change, particularly in countries such as Spain, Italy, Greece and India, production has decreased over the years, requiring new strategies to overcome these problems [1,11].

High quality and yield are now possible through corms’ cultivation in controlled environmental conditions inside plastic tunnels, greenhouses and growth chambers in hydroponic systems [1,11,12,13,14,15]. Several studies have been already conducted on the optimization of parameters, such as nutrient solution composition and fertilizer [16,17,18], corm dimension and planting density [9,19,20], substrate (perlite, vermiculite, coco peat, peat moss, Bulbfust pin-tray, etc.) [19,21,22] and forcing methods to extend flowering period of corms and increase production by controlling temperature, humidity and illumination [13,22,23,24]. Among all the environmental factors, light seems fundamental, not only for plant growth and development modulation, but also to improve the nutritional quality and yields of plant products [19,25,26,27,28,29]. Thanks to the advent of LED light technology, it is now possible to modulate light spectrum, intensity, duration and direction of the light beam, to which the plant reacts with changes in leaf morphology, adjustments in the chloroplast architecture and/or in the electron transport chain influencing photosynthesis [29]. In the light spectrum each wavelength modulates plants’ specific responses. The action of red radiation stimulates photosynthesis and the synthesis and accumulation of carbohydrates [30]; in the case of *C. sativus*, an effect on the production and higher quality of the spice has been demonstrated, increasing the size and number of stigmas [2] and regulating the activity of corm amylase and indole-3-acetic acid (IAA) [26]. Blue radiation acts on the secondary metabolism of plants, favoring the accumulation of phenylpropanoid compounds associated with antioxidant capacity [28,31,32,33,34] and, in particular in saffron, increases the weight of offspring corms by reducing their number and altering biomass distribution in favor of corms and flowers [28]. Far-red photons (above 700 nm) have minimal photosynthetic activity when applied alone, but induce photomorphogenic responses, favoring the accumulations and translocation of carbohydrates from leaves to bulb in onions and garlic [27,35], while in saffron a low red/far-red ratio during the development of offspring corms activates a phytochrome-mediated response, preparing the corms to produce flowers with a higher concentration of crocin in the stigmas [27].

Today, increasing saffron yields and quality is possible thanks to the advent of vertical farming techniques using high-density planting multilevel systems integrated with LED lights [36,37]. To produce saffron for pharmaceutical use, it is fundamental to develop a highly reliable production process inside a confined environment following both Good Agricultural Practices and Good Manufacturing Practice to guarantee a safe and stable product, preventing possible contaminations and production losses while increasing profitability [27,38,39].

On the basis of a careful analysis of current literature on hydroponic systems for the production of saffron (see Table 1) and our expertise in the area of protected cultivation, we have developed a fully automated multilevel hydroponic system, using Ebb and Drain irrigation integrated with sensors and different LED light sources for saffron farming inside sterile growth chambers and containment greenhouse.

The aim of this study was to develop a highly reliable saffron production process, through a high-density, multilevel hydroponic cultivation system within a controlled environment. An accurate study of the combined effects of different LED lighting recipes and environmental parameters was also conducted to promote flowering, optimizing yields and ensuring high quality production, free of contaminants and suitable for therapeutic applications.

## 2. Results

### 2.1. Agronomical Evaluations on the Hydroponic Cultivation Conditions

Corms of saffron were cultivated under different conditions to evaluate the effect on flowering and overall yield. Two independent production cycles were conducted over two subsequent years (2019 and 2020), using corms grown in the field by the same farm. To ensure the comparability of the results and correctly discriminate among the set experimental variables, corms with homogeneous characteristics were selected for all the experimental trials. Corms’ uniformity was evaluated by one-way ANOVA, which showed non-significant differences among the planted corms, in terms of weight, thickness, and diameter (major and minor axis of the corm) in the years 2019 and 2020 (Figure 1). 

A phase of preconditioning of the corms in the dark at a temperature of 25 °C was applied to all the experimental trials, in accordance with the hydroponic cultivation protocols of saffron reported in the scientific literature review in Table 1. Even the cultivation system, with Ebb and Drain type irrigation, was kept homogeneous among all the trials, the only exception being the trial S3_2019_ conceived for expanded clay instead of the Root It trays adopted as corm support in all the other trials.

Flowering of saffron corms was achieved under all experimental conditions. The first cultivation cycle allowed the identification of the S1_2019_ crop trial as the one that gave the best results in terms of fresh and dry weight of stigmas per flower (Table 2). However, among S1_2019_, S2_2019_ and S3_2019_ trials no relevant differences were found as to the amount of saffron produced and the yield per unit of cultivated area.

**Table 1 molecules-27-08972-t001:** Cultivation conditions of saffron in a controlled environment reported in the literature and evaluated for the setting adopted for this work, showed in the last line, highlighted in gray.

Hydroponic/ Aeroponic System	Environment	Controlled Parameters	Substrate Type	Nutrient Solution	Temperature of Flowering Induction	Light			Preconditioning/Incubation	Corms/m^2^	Flowering DAP	Flower/Corm	Flowering %	Stigma Dry Weight (mg)/ Flower	References
						Type	Photoperiod L/D	Intensity							
NFT, NMB (AE)	CGR	temperature, light, photoperiod, intensity	soil, mesh pots + bulb planting soil	½ H	17 °C ± 2	F	11/13 h	150 ± 20	no	d.n.r.	d.n.r.	d.n.r.	8–12%	4.4–4.95	[14]
WT	CGR	temperature, humidity, light, photoperiod, intensity	rockwool mat + 5 cm expanded clay (arlite)	¼ H	17 °C	F	8/16 h	20	dark, 25 °C, 85% RH, 400 ppm CO_2_	457	101–130	1–2.3	d.n.r.	10.2–10.3	[24,40]
DR	CGR; GR	temperature, humidity, irrigation	perlite, peat + perlite 1:1, perlite + vermiculite	½ H	13–31 °C GR; 17 °C CGR	S, F	8/16 h CGR	25	dark 80 days (CGR&GR) CGR 25 °C, 85% RH, 400 ppm CO_2_	50	93–111	3.8–5.4	d.n.r.	8.9–11	[13]
WT	GR; F	temperature	univ. substr. + sand (70:30 *v*/*v*) in a plastic tray	d.n.r.	17 °C GR	S	d.n.r.	200–1000	d.n.r.	320	d.n.r.	d.n.r.	d.n.r.	d.n.r.	[41]
WT	CGR; F	temperature, light	peat in a plastic tray	d.n.r.	17 °C CGR	F	8/16 h CGR	20	59–101 days, 23–27 °C, 80% RH	320–560	51–67	1.08–3.01	d.n.r.	6.57 F; 10.28 CGR	[42]
MI	PT	d.n.r.	soil + sand + manure 1:1:1, cocopite (coconut: agriperlite)	H	d.n.r.	S	d.n.r.	d.n.r.	d.n.r.	50, 100, 150	d.n.r.	1.8–2.2	d.n.r.	d.n.r.	[43]
WT-DWC	CGR	d.n.r.	d.n.r.	½ OAT A	18/15 °C; 17/17 °C	d.n.r.	8/16, 10/14	225, 162, 151, 250	90 days, 25 °C	d.n.r.	d.n.r.	d.n.r.	d.n.r.	7.2	[27]
d.n.r.	CGR	temperature, light, humidity	plastic crate	d.n.r.	d.n.r.	F, L	12/12	50	0–30 days, 25/20 °C; 31–60 days, 19/16 °C; 65–70% RH	d.n.r.	d.n.r.	d.n.r.	d.n.r.	d.n.r.	[26]
d.n.r.	GR-CGR	temperature, humidity, light	spikes tray, hydroponic tray, peat substrate	d.n.r.	d.n.r.	HPS, L	8/16	160	17.8 °C, 57.8% RH	d.n.r.	d.n.r.	2–9	d.n.r.	d.n.r.	[22]
**ED**	**CGR**	**temperature, humidity, light**	**PlantIT,** **agriperlite, greenfelt**	**IDRON**	**17 °C**	**F, L**	**8/16, 11/13 CGR**	**91, 62, 65, 215, 230**	**dark 70–105 days CGR 25 °C, 60% RH**	**400**	**4–43**	**1.10–1.48**	**16.25–63.87**	**9.05–12.81**	**-**

**Hydroponic/aeroponic system** AE: Aeroponics; NFT: Nutrient film technique; NMB: Nutrient Mist Bioreactor (aeroponic); DR Drip irrigation; WT watering; MI: Manual irrigation; DWC: Deep Water Culture; ED: Ebb and Drain. **Environment** CGR: Controlled Grow Room; GR: Greenhouse; PT: Plastic tunnel; F: Field. **Nutrient solution** H: Hoagland. **Type of Light** F: fluorescent; L: LED Light; S: Sun; HPS: High Pressure Sodium lamp. **Light intensity** in μmoles/m^2^·s. **RH**: relative humidity. **DAP**: Days after planting. **D.n.r.**: data not reported in the reference.

**Table 2 molecules-27-08972-t002:** Hydroponic cultivation outputs.

Trial i.d.	Total Planted Corm	Flowering Time (Dap) *	Total Flowers	Flowering Corm s (%)	Stigma Fresh Weight (mg)/ Flower Ratio	Total Stigma Dry Weight (g)	Flowers/Flowered Corm Ratio	Stigma Dry Weight (mg)/ Flower Ratio	Yield (g/m^2^)
**S1_2019_**	396	41	270	54.64	64.49	3.458	1.25	12.81	3.10
**S2_2019_**	396	44	319	62.91	54.83	3.509	1.28	11.00	3.20
**S3_2019_**	434	8	338	63.87	52.15	3.525	1.22	10.43	3.20
**C1_2019_**	240	7	48	16.25	42.44	0.434	1.23	9.05	0.60
**C2_2019_**	240	8	34	12.92	51.32	0.337	1.10	9.92	0.50
**C3_2019_**	240	7	123	37.91	47.98	1.194	1.35	9.71	1.80
**C4_2019_**	240	8	73	22.50	50.45	0.736	1.35	10.09	1.10
**S1_2020_**	399	4	98	19.30	61.33	1.242	1.27	12.67	1.10
**S1-T_2020_**	399	4	202	37.84	61.37	2.448	1.34	12.12	2.20
**C4-I_2020_**	399	4	186	31.58	42.98	1.875	1.48	10.08	1.70

* dap: days after planting.

The production yield obtained from the trials in the clean room (C1_2019_, C2_2019_, C3_2019_ and C4_2019_) was generally lower when compared to the trials carried out in the containment greenhouse (S1_2019_, S2_2019_ and S3_2019_) (Figure 2 and Figure 3 and Table 2). This evidence was attributed to the effect of a more frequent irrigation setting (every other day, instead of twice a week; Table 3) for the trials in the clean room, which excessively soaked the corms, likely reducing their germination capacity. However, a higher saffron production among the trials in the clean room was observed for C3_2019_ and C4_2019_, probably due to the higher intensities of blue LED light, as stated in the work of Moradi et al. [28].

The Tukey pairwise comparison suggests that the means are significantly different between trials S1_2019_ and S2_2019_, which changed only for the type of lamps, respectively LED and fluorescent light (Table 3). This analysis demonstrated that LED light increases the weight yield of fresh and dry stigmas per flower (Figure 3E,F and Appendix A).

Moreover, comparison between results of trial C1_2019_ and C3_2019_ allowed the conclusion that the duration of the photoperiod positively affects saffron production. Specifically, a shorter duration of the lighting phase (8 h in C1_2019_ vs. 11 h in C2_2019_; Table 3) correlated with a statistically significant difference in the yield ratio (dry and fresh stigma weight/flower; Figure 3 and Appendix A). 

Trial S3_2019_ was introduced specifically to evaluate the possible effect of the growing substrate on saffron production. The results demonstrated a high yield even on expanded clay (Table 2), comparable to that on the Root It tray used in all trials. However, biochemical assays showed a lower saffron quality if compared to S1_2019_. In addition, random distribution of the corms planted on the clay substrate did not allow for the application of statistical analysis to the production results.

Since the purpose of this work is not limited to the definition of an efficient soilless cultivation system but is specifically aimed at obtaining high-quality saffron intended for the drug supply chain, the identification of the optimal conditions was linked above all to the results of the biochemical assays conducted on the final product of each individual cultivation trial. In particular, the spectrophotometric analysis carried out on saffron produced in the 2019 cycle (see paragraph 2.2) allowed the S1_2019_ and C4_2019_ trials to be identified as the experimental conditions resulting in the best saffron quality in terms of crocins, picrocrocrocin and safranal.

In consideration of the results of yield and quality obtained after the first year of experimental cultivation, a second cultivation cycle was set up in the following year, to verify the reproducibility of the methodology identified as the best-performing and to evaluate the possibility of further refinement of saffron production through minor modifications. In particular, the effect of a reduction of the thermal difference between day and night in the S1_2019_ trial was investigated, lowering the daytime temperature from 25 °C to 20 °C, closer to field conditions. All other culture parameters were kept unchanged compared to the S1_2019_ trial (Table 3). The cultivation trial thus modified was named S1-T_2020_. In addition, in the new experimentation the irrigation was reduced, to tackle flooding problems leading to reduced yields in the clean room trials in the first experimental year. Furthermore, in the hypothesis that the constant temperature at 17 °C could have lowered the production capacity of the trials in the clean room, in the second experimental cycle the temperature of the C4-I_2020_ trial was also modified, with a day/night variation of 25/17 °C, similarly to the S1_2020_ and S1-T_2020_ trials. In this case as well, other parameters remained unchanged compared to trial C4_2019_. The S1_2019_ trial was replicated with the previous setting in the second cultivation cycle (and renamed S1_2020_), as an internal control.

Table 2 summarizes the results obtained from all the trials in the experimental design adopted. To evaluate the performance of *C. sativus* in the different hydroponic conditions, the data on percentage of flowered corms, the number of flowers per corm, fresh and dried weight of stigmas per flower were statistically analyzed for each trial (Figure 3). In the 2020 experiments fewer flowers were found compared to 2019, independently from the trial condition, probably attributable to the natural variability of field-derived corm supply, which in turn is affected by the seasonal trend. Despite this, the number of flowers obtained per flowering corm in 2020 remained in general comparable with that obtained in 2019, as did fresh and dried weight of stigmas.

Moreover, a punctual analysis allowed a direct comparison between the S1-T_2020_ and S1_2020_ trials, which differed only in the daytime temperature. In particular there was not a statistically significant effect on the dry weight/flower ratio due to temperature modification (Figure 3F). Instead, the lower temperature in S1-T_2020_ favored a higher flowering on the total corms planted, doubling numbers obtained in trial S1_2020_ (Figure 3B and Table 2).

In general, the one-way ANOVA highlighted only slightly significant differences (in the 2019 trials) or non-significant differences (in the 2020 trials) for the flower/corm ratio (Appendix A). Even the variations in stigma weight/flower, although statistically present, are limited (Figure 3E,F and Appendix A). More relevant is the effect of the light-temperature regime, under the conditions adopted, on the number of flowers/planted corms (Figure 3A,B) and consequently on saffron production (Figure 3C,D).

The flowering time was rather influenced by corm preconditioning in the dark at 25 °C, to which the corms are subjected from harvest to planting, than by the different cultivation conditions. In fact, the blooming occurred approximately 16 weeks after the start of preconditioning, regardless of when they began to be imbibed in the growing system. The flowering trend in the two experimental cycles (2019 and 2020) is graphically represented in Figure 2. The production in 2020 highlights the marked increase in productivity associated with S1-T_2020_ trial, while the yield of C4-I_2020_ trial is comparable to the reference S1_2020_ trial. 

### 2.2. Biochemical Characterization of Saffron Produced in a Controlled Hydroponic System

Saffron samples obtained from hydroponic cultivation under the different experimental conditions were analysed using different analytical methods, in order to characterize their chemical composition, in comparison with a sample of saffron produced in the field according to the traditional agronomical method. 

In order to have a comprehensive characterization framework of the bioactive metabolites present in the samples produced, an analysis by nuclear magnetic resonance (NMR) was carried out on deuterated methanol extracts of samples from the first production cycle (2019). The resulting spectra were dominated by the crocins and picrocrocin resonances. In addition, the resonances of two kaempferol compounds were isolated and quantified. Unfortunately, given the complexity of the mixture, it was not possible to specifically identify the two compounds, which can differ in the composition and structure of the aliphatic moieties. 

Figure 4 shows the concentration of the main components of the saffron samples collected in the trials of the 2019 crop cycle. As a general observation, the samples grown in greenhouse contained a concentration of active compounds tending to be higher than that of the samples cultivated in the cleanroom. In addition, the greenhouse samples were more similar to the sample cultivated in the field. In particular, samples S1_2019_ and S3_2019_ showed the highest concentration of metabolites, which result was even better than the control. 

To compare the bioactive compound content of the selected sample S1 under the different cultivation conditions we analyzed separately several stigmas, in order to estimate also natural variability. The average crocin and picrocrocin amounts quantified by NMR, with associated 95% confidence limits calculated by Student’s t-distribution, are reported in Figure 5. The samples collected in 2020 (S1_2020_ and S1-T_2020_) have a significative higher content of active compound compared to the 2019 (S1_2019_) samples. Field sample (Cntr) concentrations, which were evaluated by a single NMR determination on a pooled saffron sample, thus with no confidence intervals possibly estimated, are shown in the plot just as a reference. 

For a more targeted characterization of crocins, known for their effectiveness in the treatment of neurodegenerative eye diseases, a spectrophotometric analysis was carried out on the aqueous extract of saffron samples, according to ISO 3632 standard [44]. The adopted methodology allows a class for each sample and consequently its market value to be determined. 

The product class of each sample was determined following the ISO standard. It should be noticed that values refer to dry weight of saffron; therefore, another parameter to be determined is moisture. Since the amount of saffron obtained from trials S3_2019_, C1_2019_, C2_2019_, C3_2019_ trials was less than 300 mg, it was not possible to determine for these samples the three parameters required by the ISO standard. 

The absorption spectra of the samples are reported in Figure 6. It is possible to notice that the absolute maximum of the spectra (absorbance at 440 nm) was higher or equal to 1, indicating that all samples were classified as class I, in accordance with the ISO standard.

The measured absorbance values confirmed what was predicted by the spectra: (i) all the samples were of excellent quality (class I), and (ii) a great homogeneity was observed in the values of humidity, implying a very reproducible drying process of stigmas. 

The spectrophotometric analysis performed on the samples from the first cultivation cycle allowed the S1_2019_ sample to be identified as the richest in crocins, picrocrocin and safranal. Therefore, this sample was subjected to further characterization by chromatography in order to determine the concentration of trans-4 gentobiose-gentobiose crocin and trans-3 gentobiose-glucose crocin, as metabolites known for their bioactive properties [45]. Based on the concentration values of these two crocins, samples can be classified as REPRON^®^ or not. The REPRON^®^ classification certifies the neuroprotective activity of the saffron sample and consequently a possible use in the ophthalmic field. The same analysis was repeated for the S1_2020_ sample (Figure 7). The results showed that the saffron derived from the S1 cultivation conditions, in both 2019 and 2020 production cycles, meets the REPRON^®^ requirements. 

For a finer characterization of the bioactive component, an alcoholic extract of the S1_2019_ sample was also subjected to a further analysis by high-performance liquid chromatography–photodiode array detection–high-resolution mass spectrometry (HPLC-PDA-HRMS). The result confirmed that *trans*-crocetin bis (β-D-gentiobiosyl) ester (T1), also named *trans*-crocin 4, was the most abundant crocin, followed by *trans*-crocetin (β-D-gentiobiosyl) (β-D-glucosyl) ester (T2), also named *trans*-crocin 3 (Figure 8). In addition, the analysis showed the presence of crocins with a lower degree of glycosylation, considered as precursors of T1 and T2, such as *trans-*crocin 2′ (glucose-glucose crocin), *trans-*crocin 2 (gentiobiose crocin) and *trans-*crocin 1 (glucose crocin) [46]. Low levels of *trans*-crocin 5 (tri-glucose crocin) were also detected.

To summarize, the results obtained allowed us to identify optimized cultivation conditions to produce pharma-grade saffron. This hydroponic system was based on the use of Root IT trays, on an agri-perlite layer and a green felt pad. The best-performing LED lighting spectrum was composed as follows: red 75 µmol∙m^−2^∙s^−1^, blue 5 µmol∙m^−2^∙s^−1^, white 16 µmol∙m^−2^∙s^−1^; with a photoperiod of 8 h day/16 h night. The temperature leading to the best results was 20/17 °C day/night, with 60% relative humidity, irrigating twice a day with an interval of 30 min, repeated twice a week. 

## 3. Discussion

Hydroponic cultivation in a controlled environment offers a number of potential benefits: obtaining fresh plant products virtually everywhere, optimizing yields through the regulation of environmental parameters, and standardizing procedures to obtain products of certifiable quality and free from chemical treatments and contamination [47]. These conditions, meeting the major requirements for the production of plants suitable for medical uses, together with the intrinsic bioactive properties and the high added value of saffron, make this species an optimal candidate for soilless cultivation. Several attempts have been made to pursue hydroponic cultivation of saffron for food use [2,11,13,14]. Moreover, several studies demonstrated that the accumulation of saffron bioactive molecules (substantially crocins and picrocrocins) depends not only on the cultivar, but also on the cultivation environment and agronomic practices [18,20,27,30]. 

The innovative objective of this work, with respect to the present background on soilless saffron cultivation (see Table 1), was to develop a fully automated cultivation system in a sterile controlled environment that can combine high yields with the production of pharmaceutical-grade saffron. For this purpose, various experimental cultivation recipes were applied, also aimed at optimizing the concentration of molecules with therapeutic activity in the saffron produced.

In order to identify the most suitable cultivation method for the production of pharma-grade saffron, the experimental design was organized in two successive cycles. The experimentation conducted in the first cycle made it possible to identify the cultivation conditions, thus providing the best results both in terms of yield and product quality, with particular reference to therapeutic application. The best-performing cultivation recipe (namely S1_2019_) was exactly replicated in the second cultivation cycle (namely S1_2020_), to have a reference that would allow the comparison of two production years. Furthermore, in the second year, two more trials were introduced with modified parameters, in an attempt to overcome a few technical hurdles and to further improve production. The statistical analyses were carried out among the trials of each production cycle, in consideration of the registered natural variability between the two corm supplies of different years, even if they came from the same production farm. A similar experimental approach has also been adopted for saffron cultivation in other works [20]. The hydroponic system adopted was of the Ebb and Drain fully automated type to ensure the correct degree of corm imbibition and to reduce the possibility of contamination thanks to the subirrigation process.

More specifically, during the first production cycle, a series of parameters and technical solutions (growth supports, irrigation, light spectra and intensity, temperature, photoperiod) were tested and developed, setting seven cultivation trials in two independent contained environments: a class 2 greenhouse and a clean room. 

As a first evaluation of each recipe, quantitative yields were calculated, considering different flowering indices. The highest dry stigma production obtained during the whole experiment was 3.2 g/m^2^. Considering an average yield from field cultivation of 0.2 g/m^2^ [6], the production of saffron in the best hydroponic conditions adopted in this work is definitely appreciable, being up to 16 times higher than the production with traditional agronomic methods. Furthermore, it is likely to hypothesize that this yield could be further optimized, using corm lines well adapted to hydroponic cultivation, rather than novel corms coming from field. 

It was observed that the experimental trials in the greenhouse gave higher yields than those in the clean room. This result is possibly attributable to the day/night temperature difference set in the greenhouse (25/17 °C day/night), while the temperature in the clean room was kept constant at 17 °C. In fact, although at this temperature *C. sativus* still manages to bloom in the field, temperatures a few degrees higher favor the flowering process [24]. An important role was also hypothesized for irrigation, more frequent for trials in the clean room than for those in the greenhouse. The influence of the amount of water on production yields, as well as on the concentration of bioactive components, is well documented [48].

In terms of product quality, the effect of the different cultivation conditions on saffron obtained in the first experimental cycle was investigated as well. Different analyses were performed to qualify the content of bioactive compounds of saffron derived from each cultivation trial. A primary characterization method was through Nuclear Magnetic Resonance (NMR), which can provide information on the global composition of the samples. This analysis showed that the saffron samples with the highest yield corresponded to those with the highest content of bioactive molecules, with reference to crocins and picrocrocins. These findings are in agreement with the evidence reported in the literature that saffron derived from plants grown in optimal conditions can give higher yields, associated with a high quality of the spice [18,19,20].

As to the production of pharma-grade saffron destined for therapeutic applications, with particular reference to the treatment of Age-related Macular Degeneration (AMD), previous studies based on chemical qualitative and quantitative analysis, carried out in parallel with the treatment of animal models, showed that the neuroprotective activity of this spice depends on the content of active compounds. In particular, the therapeutic efficacy of saffron relies on the concentration of the two most abundant crocins: trans-crocetin bis (β-D-gentiobiosyl) ester (T1) and trans-crocetin (β-D-gentiobiosyl) (β-D-glucosyl) ester (T2). Based on these results a patented procedure (International Deposit No. W02015/145316) has been filed, to which the quality mark REPRON^®^ has been associated to saffron undergoing this procedure [3]. This procedure when applied to saffron derived from experimental trials pointed out that all samples belonged to class I, thus evidencing that hydroponic cultivation in all the experimental conditions adopted ensure an adequate concentration of active metabolites and consequently validated the product for use in the ophthalmic field. Among the products obtained in the first experimental cycle (year 2019), the spectrophotometric and HPLC analyses revealed that the S1_2019_ saffron extract presents the highest content of highly glycosylated crocins (T1 and T2), followed by the C4_2019_. This result was confirmed by sample characterization through mass spectrometry, which also highlighted a minor content of crocins with one or two glucose moieties. Crocin 5 (triglucose crocin) was also detected. The abundance of highly glycosylated crocins is correlated to a greater biological activity of the extract [3,49]. In fact, glycosylation improves water solubility of metabolites, which may be related to their improved stability and bioavailability, leading to a higher biological activity. 

Based on the results obtained in the first cultivation cycle, a second cycle was set up to validate the best-performing cultivation recipe and to verify further room for improvement. This second experimental cycle showed that the reduction of the day temperature from 25 to 20 °C was a critical factor in improving both yield and product quality. This result is in agreement with the study by Molina et al. [24] showing that the emergence of flowers is activated already at 20 °C and, although the optimal flowering range is in the range 23–27 °C (with a preferential temperature of 23 °C), prolonged exposure to high temperatures produces a detrimental effect. 

In general, a role of light on saffron flowering was observed, albeit to a lesser extent than other factors such as temperature and irrigation regime. In particular, among the trials conducted in the clean room in 2019, C3_2019_ and C4_2019_, with a higher red and blue LED component than C1_2019_ and C2_2019_, gave higher yields (Figure 3A,C; Table 2 and Table 3). This evidence was also confirmed in the 2020 cultivation cycle, in which the saffron produced for C4-I_2020_ resulted in a higher quantity than that of S1_2020_. These results are in line with what was previously described in the literature about the effect of these wavelengths on stigma production [9,28]. Nevertheless, our experimentation shows that the temperature factor plays a preponderant role compared to that played by the light spectrum composition, as S1-T_2020_ (at a day temperature of 20 °C), gave a higher production in stigmas than both S1_2020_ and C4-I_2020_ (at a day temperature of 25 °C) (Figure 3D). On the other hand, the introduction of a far-red component in the LED lighting spectrum of the cleanroom trials in the 2019 cycle did not give the expected effect of increasing crocin concentration in stigmas [27], as evidenced by the characterization carried out by both NMR and spectrophotometry (Figure 4 and Figure 6).

In conclusion, the S1-T_2020_ cultivation recipe was identified as the one that allows a combination of both a high yield (2.2 g/m^2^) and a high concentration of crocins, fundamental properties for the development of pharmaceutical preparations valid in the conservative therapy of AMD (REPRON^®^ qualification). To establish the therapeutic efficacy of the saffron obtained by this specifically developed hydroponic cultivation, our research was completed through the characterization of the neuroprotective effect on an in vitro system of retinal pigment epithelium [Di Paolo et al., submitted within the same Molecules issue].

In addition to the advantages in terms of quantity and quality of the saffron produced, it is interesting to point out that cultivation in a controlled hydroponic environment considerably facilitates harvesting operations. In fact, the raised cultivation floor makes harvesting less difficult and faster. Furthermore, the lighting control eliminates the need to harvest at dawn, carried out in the field by plucking the stigmas manually with the flower still closed. This benefit allows operating costs, which significantly affect the final cost of the product, to be contained. 

## 4. Materials and Methods

### 4.1. Supply and Preconditioning of C. sativus Corms

*Crocus sativus* Navelli cultivar (purchased from Riccardo Federici Farm, Avezzano, Italy (cultivation area 42.117778°, 13.338611°) corms were used for two subsequent cultivation experiments: the first in 2019 and the second in 2020. Defect-free corms, measuring more than 30 mm in diameter and more than 10 g in fresh weight, were selected and used in each cultivation trial. The selected corms were then stored in the dark at 25 °C for a variable period from 10 to 16 weeks after field harvest, to verify the effect of a pre-conditioning on corm flowering performance. 

### 4.2. Hydroponic Cultivation of Saffron

At the end of preconditioning, corms were sterilized with 2% hydrogen peroxide for 10 min and then cultivated under hydroponic condition at a density of 360 corms/m^2^.

Two different growth environments were tested: a containment greenhouse and an ISO class 5 cleanroom (ISO 14644-1). For the first cultivation cycle in 2019 in the greenhouse, three trials were set-up (S1_2019_, S2_2019_, and S3_2019_). For the S1_2019_ and S2_2019_ trials (Table 3) saffron corms were grown in a Growbox (Mammoth PRO: 240 × 120 × 200 cm) equipped with a Trotec TAC 1500 air purifier, configured with F7 and G4 class pre-filters and an H13 class HEPA filter for air purification in a sealed-off area via air circulation. 

Type 1 Stal&Plast (Ringe, Denmark) trays (100 × 106 cm) were used for all the trials except S3_2019_, above which the saffron corms were cultivated inside a ROOT IT Propagator 60 (55 × 32 × 5 cm) trays, where corms could fit perfectly (Figure 9). A layer of agri-perlite was placed on the bottom of each groove and a green felt pad (Manifattura Maiano, Capalle, Florence, Italy) was placed under the ROOT IT to maintain humidity. For the S3_2019_ trial only, corms were placed directly on an expanded clay layer (Gold Label Hydrocorn) distributed on the Type 1 Stal&Plast tray with a thickness of about 5 cm. A solution of Idrofill base (K-Adriatica, Loreo, Rovigo, Italy) (NPK 10–5.23, 8% CaO, 2% MgO; microelements: B 0.01%, Cu 0.02%, Fe 0.02%, Mn 0.01%, Mo 0.001%, Zn 0.003%) was used for irrigation at a 1 g/L concentration in distilled water. Irrigation was controlled by a Gronode unit (Opengrow LDA, Viseu, Portugal), the brain of the Grolab system through the Powerbot unit (electronic device controller, Opengrow LDA, Viseu, Portugal) that activates the submersible pumps in the main tank of the Ebb and Drain system. For providing the correct quantity of water and nutrients to the corm roots, a hybrid approach with a pre-programmed water delivery regime, monitored by substrate moisture and temperature sensors (connected to the soilbot unit; Opengrow LDA, Viseu, Portugal), was used. The Ebb and Drain regimens have been tested, setting sub-irrigation twice a day, with an interval of 30 min, repeated twice a week (S1_2019_, S2_2019_, S3_2019_, S1_2020_, S1-T_2020_ and C4-I_2020_ trials), or every other day (C1_2019_, C2_2019_, C3_2019_, C4_2019_ trials). 

For the S1_2019_ trial, Lumigrow PRO 650 LED lamps were used. For the S2_2019_ trial, a StarLight Prima Klima 4 × 55 W lamp holder with two Grow 6500 K neon tubes and two TCL ProStar 2100 K neon tubes was used, while for the S3_2019_ trial Lumigrow PRO 325 lamps were used. For the S1_2019_, S2_2019_ and S3_2019_ trials, the temperature was set as a function of the photoperiod: 8 h of light at 25 °C and 16 h of dark at 17 °C (Table 3). 

In the cleanroom, four trials, namely C1_2019_, C2_2019_, C3_2019_ and C4_2019_, were set up under different light conditions with specifically water-cooled lamps realized by G&A Engineering for the IDROZAFF project, with Osram Oslon SSL LED (Far Red 730 nm; Hyper Red 660 nm; Deep Blue 450 nm; White 5000K CRI70), the light recipe being designed with the Osram Horticulture LED web tool. For the C1_2019_ and C3_2019_ trials, the photoperiod was 8 h of light and 16 of dark, while for the C2_2019_ and C4_2019_ trials, it was 11 h of light and 13 of dark. Temperature was set to be constant at 17 °C. 

For the second cultivation experiment in 2020, the best-performing trials from the 2019 experiment were replicated. The new trials, namely S1_2020_, S1-T_2020_ and C4-I_2020_, were set up, where S1_2020_ was conducted under the same conditions adopted in the 2019 experiment, S1-T_2020_ was a replica of the S1_2019_ trial where the day temperature was set at 20 °C, and C4-I_2020_ was a replica of the C4_2019_, being irrigated twice a week instead of every other day. 

All experimental trials were conducted in an environment with relative humidity control set at 60%.

Saffron production data were statistically elaborated by analysis of variance (one-way ANOVA) followed by Tukey’s multiple comparison test with Prism 9 (GraphPad Software LLC, CA, USA) program. 

### 4.3. Saffron Harvesting and Conservation

Saffron stigmas were collected during the flowering period (along 10–20 days). After recording the fresh weight, stigmas were dried in a thermobalance (Radwag MA.50/1.X2.IC.A), until they reached about 10% residual humidity. Dried stigmas were then vacuum-packed and stored at 4 °C, in the dark. 

### 4.4. Spectrophotometric Analysis

Sample preparation for spectrophotometric analysis was carried out according to the ISO-3632 procedure [44]. About 50 mg of saffron stigma were gently ground in a mortar. Ten mg of the powdered sample were suspended in a 20 mL volumetric flask filled with 18 mL of distilled water; the suspension was kept under magnetic stirring for 1 h in the dark and finally diluted to 20 mL. The spectrophotometric measurement was carried out on a suitable aliquot of aqueous extract after a 10-fold dilution and filtration on a 0.45 μm Whatman Spartan 13/0.2 RC (Whatman, GE Healthcare Life Sciences, Little Chalfont, UK) cellulose filter. The UV-visible spectra were acquired in the 200–600 nm range with a UV-30 Scan Onda spectrophotometer using a 1 cm pathway quartz cuvette and pure water for blank correction. The spectra were recorded with a 1 nm resolution. 

From the absorbance values at three different wavelengths (440, 330, 257 nm), determined from the UV-Vis absorption spectrum of the aqueous extract of saffron, three molar extinction coefficient values were measured: E440, referring to the coloring power, based on 440 nm absorption for crocin and crocetin; E257, referring to the bittering power, based on 257 nm absorption for picrocrocin; E330, referring to the aromatic power, based on 330 nm absorption for safranal. 

### 4.5. Classification by a High-Performance Liquid Chromatography-Based Method

Chromatographic analysis was performed on a hydro-alcoholic extract of the sample using Method II, previously described [50]. The percentage concentration of the two crocins, trans-4 gentobiose-gentobiose and trans-3 gentobiose-glucose, was measured in mg, with reference to 100 g of dry saffron, using the following formula [49]:(1)$$c(mg/g)=\frac{MW_{i}*E_{cm}^{1\%\;\;\;}\left(440nm\right)*\;A_{i}}{\varepsilon_{t,\;c}}$$
where $MW_{i}$ and $A_{i}$ are the molecular weight and the percentage peak area, respectively, $E_{cm}^{1\%}$ (440 nm) is the coloring strength of the saffron sample and $\varepsilon_{t,c}$ is the extinction coefficient (89.000 M^−1^ cm^−1^ for 215 trans-crocins and 63.350 M^−1^ cm^−1^ for cis-crocins).

### 4.6. NMR Analysis

Saffron dried samples were stored at room temperature (RT) in an anhydrous environment protected from the light. A set of 10 samples made by a pool of three stigmas was analyzed for each cultivation condition. Stigmas were ground by tungsten beads in a 2 mL tube at maximum vortex speed for 1 min, obtaining a uniform size powder. 

About 5 mg of saffron stigma powder were suspended in 200 µL of deuterated solvent (CD3OD, DMSO-6d). The samples, protected from direct light, were sonicated for 10 min at room temperature. After 40 min of incubation at RT, the samples were sonicated again for an additional 10 min before being centrifuged for 5 min at 2000× *g*. The supernatant was then recovered and analyzed by NMR. 

1H NMR spectra of saffron samples were recorded on a Bruker 600 AVANCE spectrometer (Bruker, Billerica, MA, USA) operating at 600.13 MHz at 298 K. The spectra were collected with a 45° pulse of 5.0 µs and a relaxation delay of 2 s. The residual HOD signal was suppressed by presaturation during the relaxation delay. The NMR signal was Fourier-transformed and phase- and baseline-corrected. 

Relative concentrations of the metabolites of interest were estimated by peak integration. Averages and confidence intervals are calculated by Python Scipy Library.

### 4.7. High-Resolution Mass Spectrometry Analysis

A hydroalcoholic extract of saffron sample S12019 was analyzed by high-performance liquid chromatography–photodiode array detection–high-resolution mass spectrometry (HPLC-PDA-HRMS), using a Q-Exactive mass spectrometer (Thermo Fisher Scientific, Waltham, MA, USA) equipped with a photodiode array detector (Dionex, Sunnyvale, CA, USA). Briefly, the *C. sativus* stigma hydroalcoholic extract was prepared as follows: 3 mg of *C. sativus* stigmas were pulverized by agitation for 2 min at 20 Hz frequency in a mixer mill MM 300 (Retsch) in the presence of 2 mm steel beads; the powder was resuspended in 300 µL of cold 50% (*v*/*v*) methanol, homogenized for 40 min in MM 300 at 20-Hz frequency, and centrifuged for 20 min at 20,000× *g*. The supernatant was recovered, and 2 µL of it were subjected to HPLC-PDA-HRMS analysis. HPLC separation, electrospray ionization and metabolite identification were performed as previously described [46,51]. All solvents used were LC-MS grade (Merck Millipore). 

## 5. Conclusions

This work allowed the devising of a hydroponic cultivation system for saffron, in a controlled and clean environment, suitable for the production of pharma-grade saffron. The adoption of high-tech agronomic solutions has led to a standardized procedure, not feasible in field conditions, to produce high-quality spice, with bioactive components of certifiable quality and concentration for the treatment of neurodegenerative diseases of the eye, such as AMD.

The yield per unit of cultivated area was particularly appreciable and higher than field production, even under the best conditions. Interestingly, cultivating in a closed and controlled environment, using multilevel growing systems, offers the possibility of increasing production with no environmental and seasonal limitations. In addition, it is possible to foresee a future optimization of the system to achieve, through appropriate corm preconditioning, doubling or tripling of production per year.

## Figures and Tables

**Figure 1 molecules-27-08972-f001:**
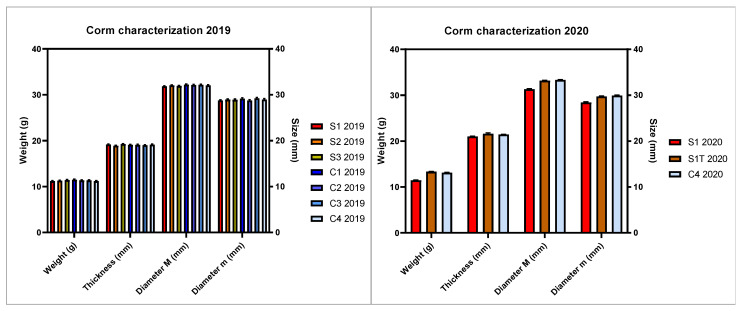
Characterization of the corms planted in the first (2019) and second (2020) experimental cycle. Bars represent means with standard error (SEM) of corm weight, thickness, diameter (major *M* and minor *m* axes). The Tukey test revealed the absence of significant differences between the means.

**Figure 2 molecules-27-08972-f002:**
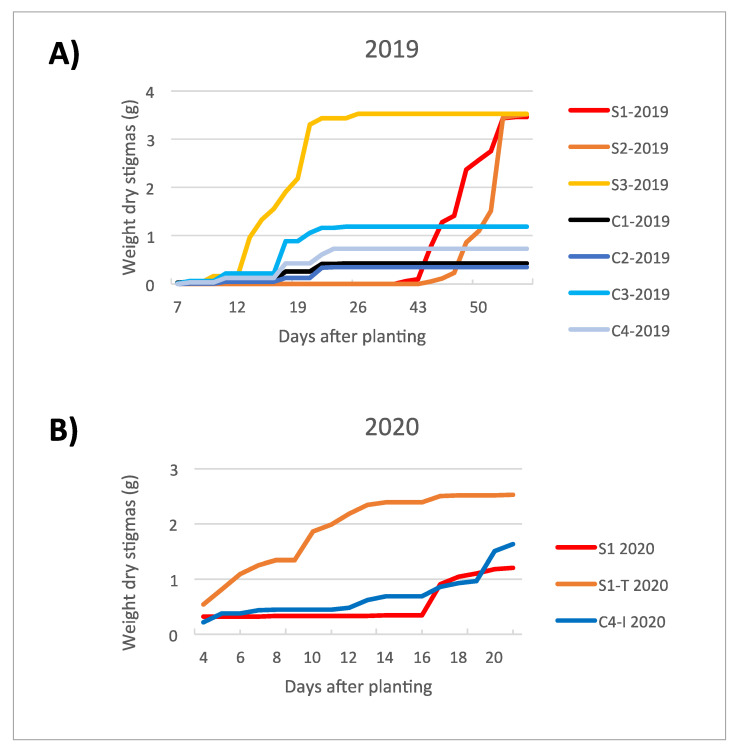
Increasing trend in stigma dry weight produced under the different environmental conditions of the trials set for the cultivation cycles in 2019 (**A**) and 2020 (**B**).

**Figure 3 molecules-27-08972-f003:**
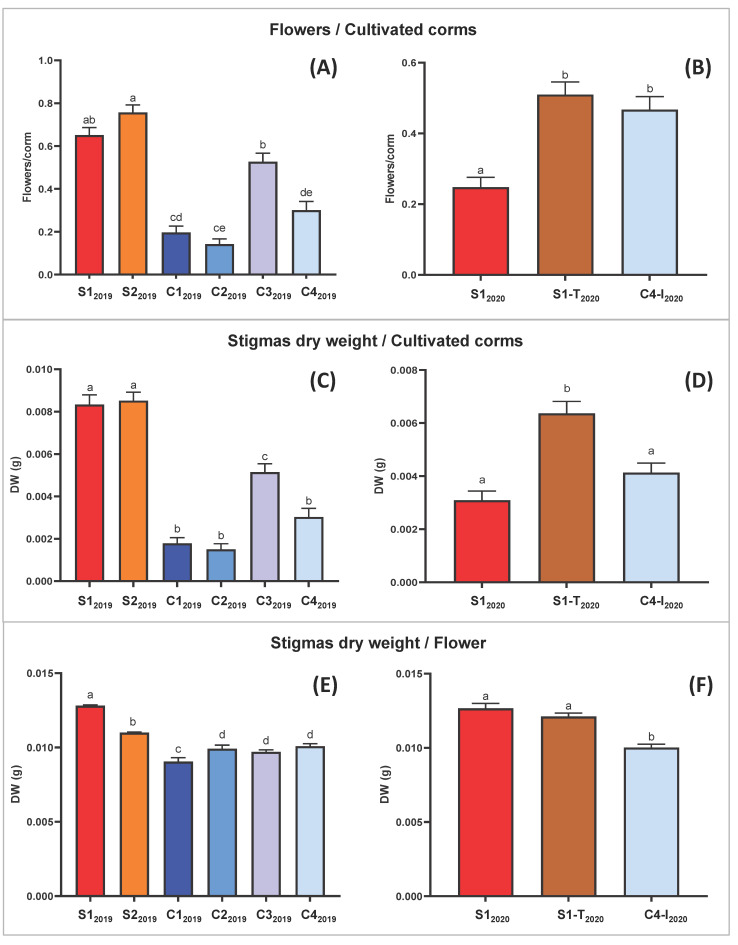
Pharma-grade saffron production data. Bars represent mean values with standard error of the mean. Mean of the ratio between flowers and cultivated corms, in the first (2019) (**A**) and in the second (2020) (**B**) experimental cycle. Mean stigma dry weight yield per corm of total corms, in the first (2019) (**C**) and in the second (2020) (**D**) experimental cycle. Mean stigma dry weight yield per flower of flowered corms, in the first (2019) (**E**) and in the second (2020) (**F**) experimental cycle. In all the graphs, one-way ANOVA results were extremely significant with *p* ≤ 0.0001. Means not sharing any letter are significantly different by the Tukey test at the 5% level of significance.

**Figure 4 molecules-27-08972-f004:**
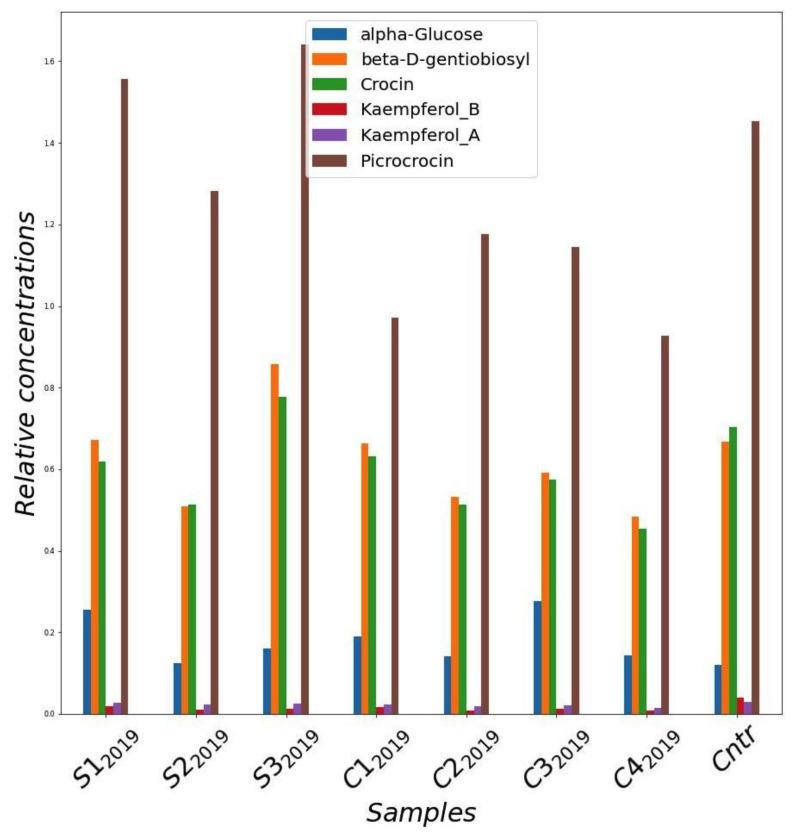
Relative concentrations of main metabolites quantified from NMR spectra of methanol extracts of saffron stigmas for the different samples harvested in 2019 year. Cntr: field saffron control.

**Figure 5 molecules-27-08972-f005:**
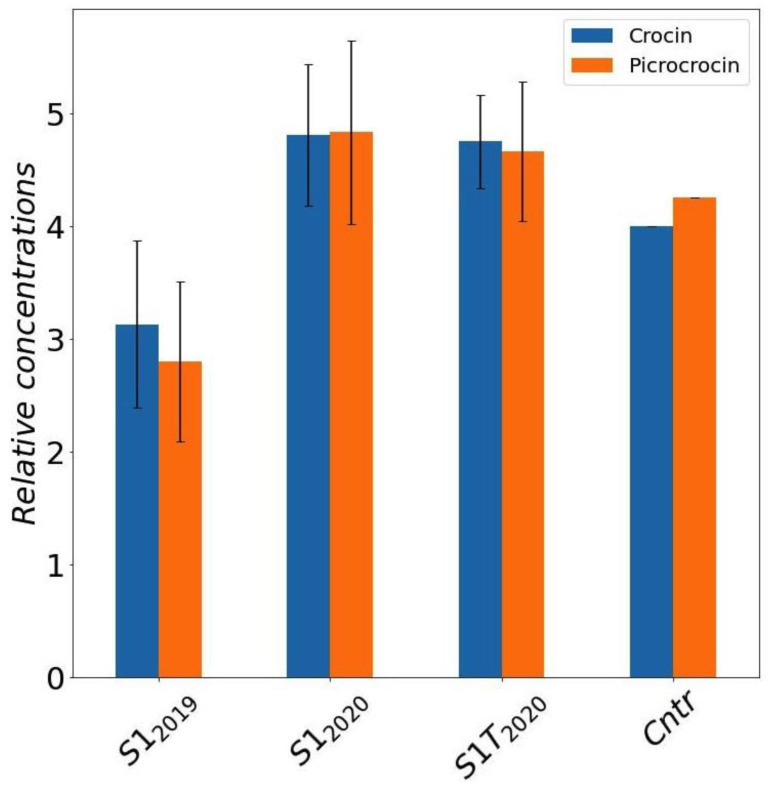
Average concentrations of crocins and picrocrocin for S1 samples grown in different conditions and seasons obtained by DMSO extracts of saffron stigmas (error bars represent the 95% confidence intervals calculated by *t*-distribution). Mean and confidence intervals are calculated over different NMR determinations in a set of stigmas coming from the same trial. Results for saffron produced in field are reported for comparison. Cntr: field saffron control.

**Figure 6 molecules-27-08972-f006:**
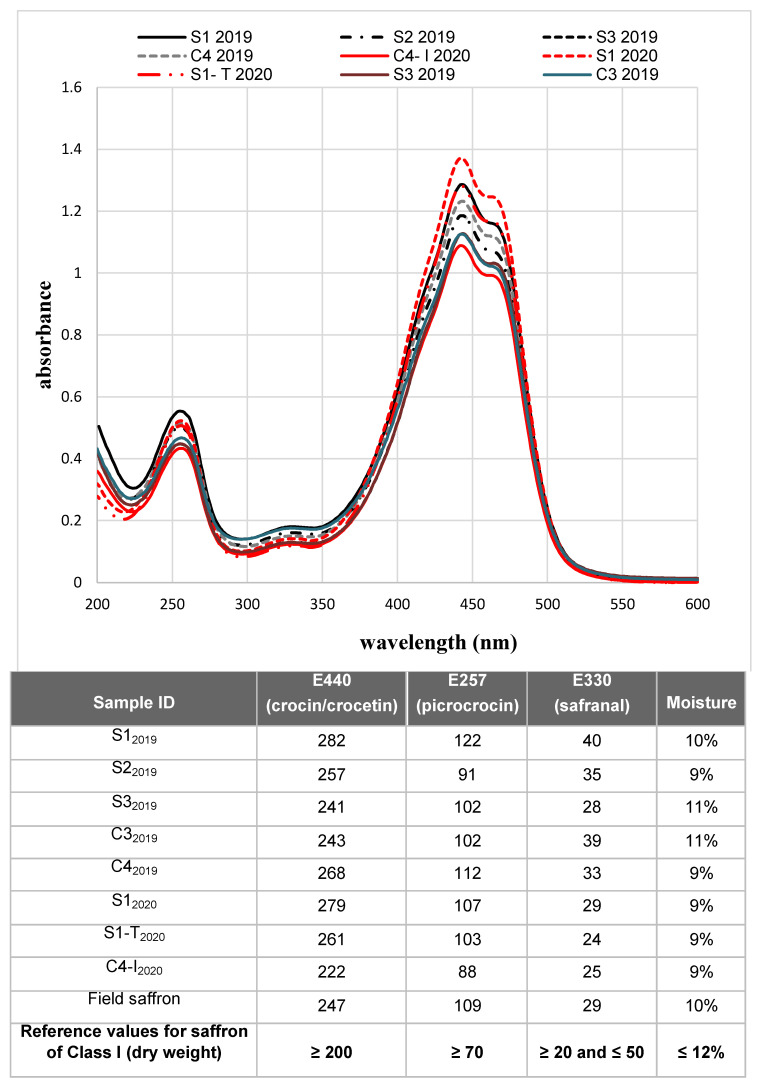
Spectrophotometric analysis of saffron samples. The absorption spectra of saffron samples produced in the two cultivation cycles are shown. For the 2019 cycle, only results for samples in sufficient quantity and with the adequate degree of humidity for the analysis are reported. The table below details the values of coloring (E440), bittering (E257) and aromatic (E330) power of the samples for which it was possible to determine the moisture. The results obtained are compared with the reference values according to ISO 3632 standard.

**Figure 7 molecules-27-08972-f007:**
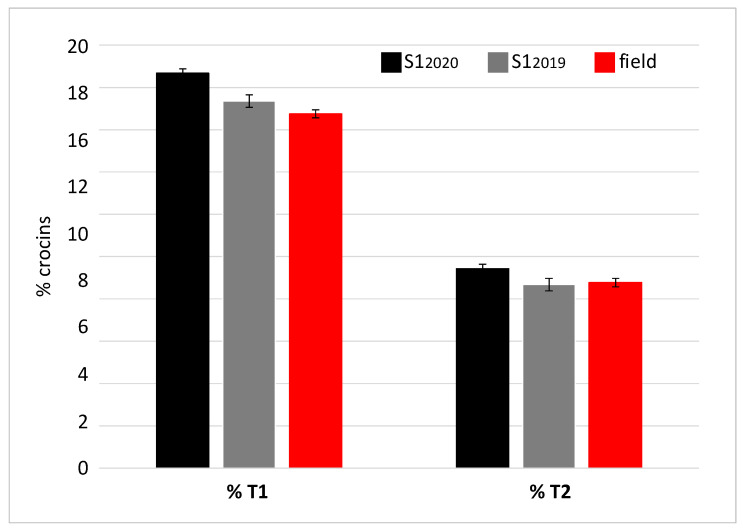
Percentage concentration (% mg), referred to 100 g of dry saffron, of the two T1 and T2 crocins, relative to the two samples S1_2019_ and S1_2020_ and field-grown saffron.

**Figure 8 molecules-27-08972-f008:**
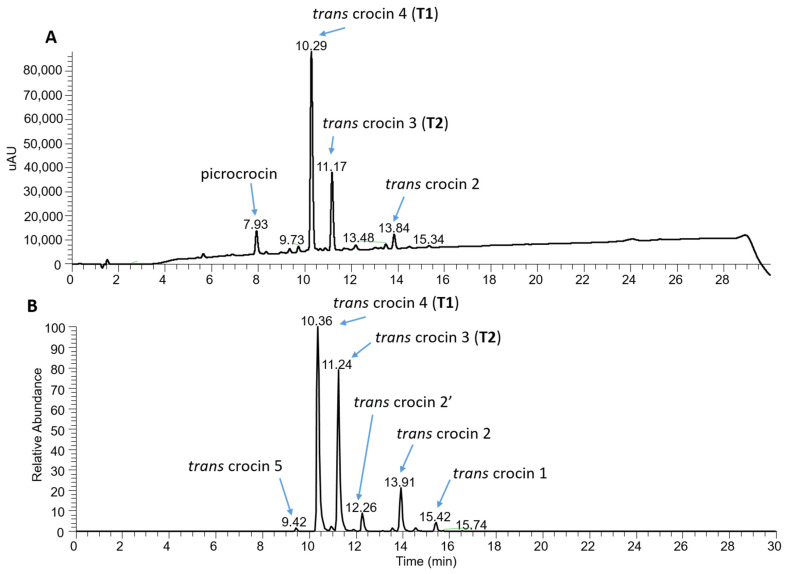
Characterization of crocin content of the S12019 sample. (**A**) PDA chromatogram (full scan absorption; 230–700 nm) showing the presence of the most abundant crocins and of picrocrocin. (**B**) Electrospray ionization +/MS chromatograms of the extracted accurate mass of crocetin (M + H + 329.1747) generated from crocin fragmentation.

**Figure 9 molecules-27-08972-f009:**
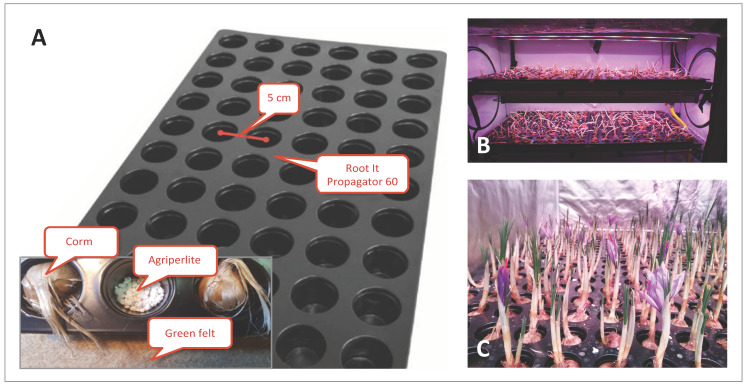
Saffron cultivation in a hydroponic system. (**A**) Assembly of the corm imbibition system, using Root It trays. (**B**) Cultivation trial conducted in Growbox. (**C**) Flowering of saffron in a controlled hydroponic environment.

**Table 3 molecules-27-08972-t003:** Hydroponic cultivation conditions.

Trial i.d.	Preconditioning	Lighting System	PFD (µmoli∙m^−2^∙s^−1^)	SDI (µmoli∙m^−2^∙s^−1^)	T_day/night_ - RH	Day/Night	Cultivation Method	Irrigation
**S1_2019_**	10 weeks at 25 °C dark	LED Lumigrow PRO 650 (x1)	91	R: 75 B: 5 W: 16	25/17 °C - 60%	8 h/16 h	green felt; ROOT IT	twice a day with a 30 min interval/ twice a week
**S2_2019_**	10 weeks at 25 °C dark	StarLight—4 × 55 W—Prima Klima (x1)	65	2x 55 W 6500 K 3800 lux/2x 55 W 2100 K 3300 lux	25/17 °C - 60%	8 h/16 h	green felt; ROOT IT	twice a day with a 30 min interval/ twice a week
**S3_2019_**	16 weeks at 25 °C dark	LED Lumigrow PRO 325 (x2)	62	R: 52 B: 8 W: 12	25/17 °C - 60%	8 h/16 h	green felt; expanded clay	twice a day with a 30 min interval/ twice a week
**C1_2019_**	15 weeks at 25 °C dark	LED G&A Engineering (x2)	215	R: 84 B: 25 W: 98 FR: 9	17/17 °C - 60%	8 h/16 h	green felt; ROOT IT	twice a day with a 30 min interval/every other day
**C2_2019_**	15 weeks at 25 °C dark	LED G&A Engineering (x2)	215	R: 85 B: 25 W: 95 FR: 10	17/17 °C - 60%	11 h/13 h	green felt; ROOT IT	twice a day with a 30 min interval/every other day
**C3_2019_**	15 weeks at 25 °C dark	LED G&A Engineering (x2)	230	R: 114 B: 41 W: 73 FR: 9	17/17 °C - 60%	8 h/16 h	green felt; ROOT IT	twice a day with a 30 min interval/every other day
**C4_2019_**	15 weeks at 25 °C dark	LED G&A Engineering (x2)	230	R: 109 B: 41 W: 71 FR: 10	17/17 °C - 60%	11 h/13 h	green felt; ROOT IT	twice a day with a 30 min interval/every other day
**S1_2020_**	14 weeks at 25 °C dark	LED Lumigrow LUMIBAR (x3)	91	R: 75 B: 5 W: 16	25/17 °C - 60%	8 h/16 h	green felt; ROOT IT	twice a day with a 30 min interval/ twice a week
**S1-T_2020_**	14 weeks at 25 °C dark	LED Lumigrow PRO 650 (x1)	91	R:75 B:5 W16	20/17 °C - 60%	8 h/16 h	green felt; ROOT IT	twice a day with a 30 min interval/ twice a week
**C4-I_2020_**	14 weeks at 25 °C dark	LED G&A Engineering (x4)	230	R:109 B:41 W:71 FR:10	25/17 °C - 60%	11 h/13 h	green felt; ROOT IT	twice a day with a 30 min interval/ twice a week

PFD: Photon Lux Density; SDI: Spectrum distribution and intensity; RH: relative humidity; Pp: Photoperiod.

## Data Availability

Not applicable.

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
