# Peer review of "Farming for Pharming: Novel Hydroponic Process in Contained Environment for Efficient Pharma-Grade Production of Saffron"

_molecules, 2022, doi:10.3390/molecules27248972_

Round 1
Reviewer 1 Report
1) Statistical analysis in not performed with anova or any other statistical analysis tools.
2) Is there any difference in the growing condition for among s1,s2,s3 and C1,C2,C3 and C4.
If can provide a table with condition will be better.
3) Humidity in the green house condition are not specified.
4) Location of sample collection need to be mention in the manuscript.
5) Age-related Macular Degeneration (AMD) is specifically mentioned in the manuscript, but there is no biological activity experiments conducted. It is mandatory to understand the anti-oxidant capacity of the product produced through the hydroponics.
6) There are many reports about the production of saffron through hydroponics technology and should be specified about the novelty of the method used here.
Reviewer 2 Report
This paper illustrated a novel hydroponic system in contained environment for efficient pharma-grade production of saffron. This paper has good operability and strong guidance. However, in the introduction part, there is a lack of induction of the existing problems and refinement of the innovation points of this article, so the innovative description of this article should be strengthened
Reviewer 3 Report
Dear Authors,
The work presented for review: “Farming for Pharming (I): Novel hydroponic system in contained environment for efficient pharma-grade production of saffron” contains interesting technological solutions and could be important for the management of horticultural practice, if it complied with all conditions of experimental work.
Detailed comments:
1) The title of the work does not raise any objections.
2) Abstract. The " cultivation protocol" provided has no statistical justification.
3) At the end of the chapter "Introduction", the Authors, in addition to the aim of the work, should put forward an alternative research hypothesis to the null hypothesis and then verify it in the text
4) In sub-section 2.1 "Assessment of agronomic conditions in hydroponic cultivation", the interpretation of the results is insufficient as there is no statistical evaluation of the test results. In addition, the data in the section "Material and methods" should be supplemented with a description of the planted tubers, what was their health class, which could have had a significant impact on the results obtained.
5) In the same subsection (2.1), irregularities were found in the conduct of the experiment. Changing the conditions of safflower cultivation during the experiment has a major impact on the obtained results; under constant experimental assumptions, this should not be the case. A new series of experiments had to be carried out,
6) In chapter 2.2, methodical matters should be moved to the chapter "Material and methods", and in this chapter only the research results should be discussed.
7) In the chapter "Discussion" the authors actually discuss their results once again and confront them little with other authors - first, secondly, the authors did not explain why during the experiment the hydroponic cultivation procedure was changed, it is mainly about changing the daily temperature from 25 at 20oC, change of lighting conditions, change of irrigation frequency in the second year of cultivation? Moreover, how do we know that it was the temperature that influenced the quality of the obtained product, and not another factor (e.g., light intensity, wavelength), since the experiments were not repeated in the next year? And no statistical calculations were made to confirm this thesis?
8) In the chapter "Material and methods", two major mistakes were made:
A. the description of the experiment shows that the assumptions of the experiment were not well thought out. The authors changed the conditions of the experiment (including air temperature, light conditions, frequency of irrigation) during the experiment (in the second year). The authors did not repeat this experiment the following year, but drew conclusions and conducted discussions based on the results obtained in the second year of hydroponic cultivation. This is unacceptable from the point of view of the principles of field or greenhouse experimentation. The experiment had to be repeated in the following year so that the results could be statistically analyzed
B. The "Statistical Calculations" section is missing from this chapter. The authors did not carry out a statistical analysis of the obtained research results, which is the basic requirement in this type of experiment. Moreover, changing the experimental conditions (e.g., changing the type of lighting, changing the temperature or changing the frequency of irrigation) during the experiment completely changes the experimental conditions and undermines the basic principles of greenhouse experiments. Such action has a significant impact on the results of the research. The results of the research obtained by the authors were non orthogonal. While there are methods for calculating non-orthogonal research results, the authors did not.
9) The conclusions presented in the paper, due to improperly conducted experiment, contrary to the rules of greenhouse experimentation and the lack of repeatability of the results, are unjustified.
Round 2
Reviewer 3 Report
Dear Authors,
The work has been supplemented and corrected in accordance with the comments. The abstract of the thesis was supplemented and corrected. The introduction was supplemented with an explanation of the purpose of the work. The chapter "Material and methods" has been supplemented and better described. The experimental part was supplemented with data on irrigation, spectrometric analysis, as well as statistical analyses, and the importance of statistical calculations in the interpretation of test results was emphasized. The biggest changes took place in the "Results" chapter. The authors tried to interpret the results of the research in more detail. The chapter "Discussion" has also been supplemented, where the authors made better use of the literature on the subject. In the "References" chapter, the authors added 4 items used in the work.
